# HALoS: Hierarchical Asynchronous Local SGD over Slow Networks for Geo-Distributed Large Language Model Training

**Geon-Woo Kim** [1]  **Junbo Li** [1]  **Shashidhar Gandham** [2]  **Omar Baldonado** [2]  **Adithya Gangidi** [2]  **Pavan Balaji** [2]
**Zhangyang Wang** [1]  **Aditya Akella** [1]

## Abstract

Training large language models (LLMs) increasingly relies on geographically distributed accelerators, causing prohibitive communication costs across regions and uneven utilization of heterogeneous hardware. We propose HALoS, a hierarchical asynchronous optimization framework that tackles these issues by introducing local parameter servers (LPSs) within each region and a global parameter server (GPS) that merges updates across regions. This hierarchical design minimizes expensive inter-region communication, reduces straggler effects, and leverages fast intra-region links. We provide a rigorous convergence analysis for HALoS under non-convex objectives, including theoretical guarantees on the role of hierarchical momentum in asynchronous training. Empirically, HALoS attains up to $7.5\times$ faster convergence than synchronous baselines in geo-distributed LLM training and improves upon existing asynchronous methods by up to $2.1\times$. Crucially, HALoS preserves the model quality of fully synchronous SGD—matching or exceeding accuracy on standard language modeling and downstream benchmarks—while substantially lowering total training time. These results demonstrate that hierarchical, server-side update accumulation and global model merging are powerful tools for scalable, efficient training of new-era LLMs in heterogeneous, geo-distributed environments.

## 1. Introduction

Large Language Models (LLMs) have rapidly revolutionized diverse domains (Dubey et al., 2024; Achiam et al., 2023), but their training presents significant computational challenges. Today's de facto standard distributed optimization approach to pretrain LLMs is fully synchronous Stochastic Gradient Descent (SGD), where the massive amount of model states are synchronized across multiple accelerators (e.g., GPUs and TPUs) after every training step. To overcome this communication overhead, practitioners attempt to deploy homogeneous accelerators in close proximity and connect them with highly optimized network infrastructure (Dubey et al., 2024; Jiang et al., 2024). For example, to pretrain Llama-3 405B, 16K H100 GPUs are deployed within a single datacenter, each equipped with 400 Gbps network bandwidth and network connectivity with latency in the tens of microseconds (Dubey et al., 2024).

While a highly optimized homogeneous single-datacenter setup is desirable, it faces significant scalability challenges due to operational constraints, such as energy consumption and power management, when deploying large numbers of accelerators in one region (Gherghescu et al., 2024). Similarly, companies and researchers relying on cloud providers often struggle to allocate sufficient GPUs in a single region due to limited availability (Strati et al., 2024).

Consequently, there is a growing need to utilize accelerators that are *geo-distributed* across multiple regions in training LLMs (Yuan et al., 2022; Tang et al., 2024; Gandhi et al., 2024; Jaghouar et al., 2024; Strati et al., 2024). This setup presents new challenges: (1) inter-region communication is orders of magnitude slower, with link bandwidths typically ranging from 0.1 to 10 Gbps and network latency in milliseconds (Jaghouar et al., 2024; Gandhi et al., 2024; Yuan et al., 2022), and (2) accelerators are highly heterogeneous, as maintaining homogeneous hardware across multiple regions is impractical (Li et al., 2022; Mei et al., 2024).

Local SGD, a widely studied communication-efficient extension of synchronous SGD, allows workers[1] to perform multiple local model updates before global synchronization, balancing computation and communication overhead (Zhang et al., 2015; Stich, 2018; Lin et al., 2018). Recent efforts have adapted this approach to address the challenges in

---

[1]The University of Texas at Austin [2]Meta. Correspondence to: Aditya Akella <akella@cs.utexas.edu>.

*Proceedings of the 42$^{nd}$ International Conference on Machine Learning*, Vancouver, Canada. PMLR 267, 2025. Copyright 2025 by the author(s).

[1]In this paper, a worker represents a set of accelerators functioning as a data-parallel group, maintaining a model replica.

geo-distributed LLM training. DiLoCo (Douillard et al., 2023) demonstrates empirically that carefully tuned momentum acceleration (Sutskever et al., 2013) enables local SGD to amortize slow inter-region communication costs effectively. However, under heterogeneous accelerators, DiLoCo's strictly synchronous design often results in significant resource underutilization due to the straggler problem.

Async-Local-SGD (Liu et al., 2024) addresses this by incorporating asynchrony, allowing faster workers to proceed independently. It employs a global server that asynchronously collects and applies gradients to update the global model, while workers periodically pull the latest parameters, compute gradients, and push them back (Langford et al., 2009; Dean et al., 2012; Li et al., 2014). Async-Local-SGD enhances convergence through momentum correction mechanisms that mitigate the effects of stale gradients and the variability inherent in asynchronous training.

However, Async-Local-SGD encounters two major challenges. First, slow communication between the global server and workers hinders convergence. Although asynchrony allows workers to operate independently and mitigates the straggler problem, regular communication is still required for pulling the latest models and pushing computed gradients. The slow inter-region communication can significantly extend these times. While increasing the number of local updates helps amortize communication overhead, it increases the bias and variance of gradients in local SGD (Ortiz et al., 2021; Balles et al., 2024), degrading convergence efficiency and requiring more tokens to achieve the same model performance, offsetting the benefits of reduced communication overhead. Second, theoretical analysis and convergence guarantees for asynchronous training methods under practical geo-distributed conditions remain underexplored. While Async-Local-SGD empirically demonstrates faster convergence compared to synchronous methods, the lack of a theoretical basis limits broader adoption in practice. Key aspects, such as the impacts of communication delays and staleness in gradients, data heterogeneity, and the role of momentum, remain poorly understood.

**Our Approach.** We present a new technique for geo-distributed training of LLMs that overcomes the above challenges. It effectively controls the impact of slow inter-region communication central to the geo-distributed setup.

We develop HALoS, a hierarchical distributed optimization framework. As shown in Figure 1, HALoS deploys local parameter servers within each region that asynchronously update their local models using gradients computed by workers in the same region, leveraging fast intra-region communication. Here, each worker performs multiple local updates before communicating with their local parameter server. At the higher tier, a global parameter server coordinates training progress across local servers, hiding slow inter-region

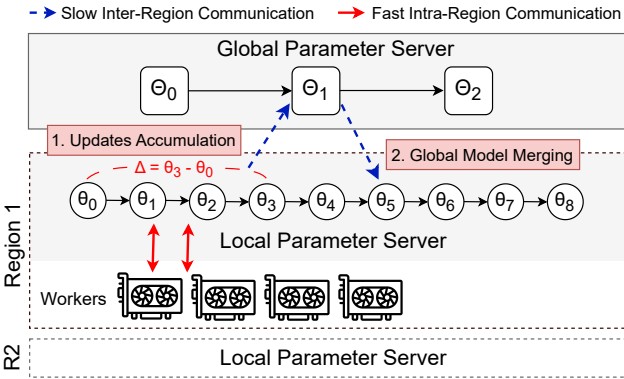

*Figure 1.* Overview of HALoS. Local parameter servers leverage fast intra-region networks (red solid lines) to update their models with asynchronously computed gradients from workers. Accumulated updates are sent to the global server via slow inter-region networks (blue dotted lines), and the latest global model is merged back into the local models.

communication through asynchronous updates. In particular, local servers continue updating their local models during communications with the global server, and merge the updated local model with the model pulled from the global server (we refer to this as "global model merging").

This novel hierarchical design of HALoS offers several benefits, ensuring efficiency and scalability, and achieving good model performance in geo-distributed training. First, the introduction of an additional level of hierarchy significantly lowers the communication stress on high-delay and low-bandwidth network links, and mitigates their impact on training performance. Second, local servers accumulate updates and communicate with the global server at optimized intervals, significantly reducing the overhead on the global server and ensuring its scalability. Third, by combining asynchronous updates and global model merging, HALoS minimizes computation idle time (compared to Async-Local-SGD) and prevents valuable updates from being discarded in the presence of the remnant high inter-region communication delays.

We provide a formal algorithm description and rigorously prove its convergence for general non-convex loss functions, with a focus on the role of momentum at each hierarchy level and the impact of delays in hierarchical asynchronous designs. Our analysis offers insights into balancing local and global updates to enhance convergence efficiency in challenging geo-distributed settings.

In sum, our contributions are as follows:

- We propose HALoS, a novel hierarchical asynchronous framework that achieves scalable, efficient geo-distributed LLM training, enabling fully asynchronous communication at each level with server-side updates accumulation and global model merging.

- We present a tight convergence proof for HALoS with general non-convex loss functions, offering insights and theoretical guarantees on the complex dynamics of asynchronous training. To the best of our knowledge, this is the first convergence proof for hierarchical asynchronous distributed optimization with momentum.

- We demonstrate that HALoS achieves up to $7.5\times$ faster convergence than DiLoCo and up to $2.1\times$ faster convergence than Async-Local-SGD in geo-distributed environments. Compared to fully synchronous SGD, HALoS achieves $68.6\times$ faster convergence and matches model performance in standard benchmarks. To foster reproducibility, we release our implementation at https://github.com/utnslab/halos.

## 2. Related Work

**Geo-distributed LLM Training.** Recent research has focused on enabling LLM training in the geo-distributed environment, where computational resources such as GPUs and TPUs are distributed across different geographical regions (Yuan et al., 2022; Strati et al., 2024; Gandhi et al., 2024; Tang et al., 2024). These approaches leverage various training parallelism strategies, including data parallelism, tensor parallelism, and pipeline parallelism (Narayanan et al., 2021), to formulate training costs and minimize them by finding the most communication-efficient and computationally balanced partitioning schemes. For example, they observe that mapping pipeline parallelism, where relatively small amount of activations are transferred, to slow inter-region communication is beneficial. However, they either do not account for heterogeneous accelerator speeds in their optimization search space (Yuan et al., 2022; Strati et al., 2024; Gandhi et al., 2024), potentially leading to suboptimal resource allocation, or they rely on gradient and activation compression techniques (Tang et al., 2024) without providing rigorous theoretical proofs or detailed analyses of their impact on training dynamics and convergence.

**Local SGD.** Local SGD enables efficient distributed optimization in communication-constrained environments, where local models are synchronized infrequently after multiple updates (Zhang et al., 2015; Stich, 2018; Lin et al., 2018; Yu et al., 2019; Wang et al., 2019; Castiglia et al., 2021; Sun et al., 2024). These methods significantly reduce communication overhead while achieving linear convergence speed-ups with more workers, matching the theoretical rates of synchronous SGD for both convex and non-convex objectives (Koloskova et al., 2020; Khaled et al., 2020; Wang & Joshi, 2021). DiLoCo (Douillard et al., 2023) and Async-Local-SGD (Liu et al., 2024) adapt local SGD for geo-distributed LLM training and show improved training efficiency over synchronous SGD with carefully selected and designed momentums. However, in geo-distributed

settings, their performance under slow inter-region communications can be limited due to the trade-off between training and communication efficiency (Balles et al., 2024; Ortiz et al., 2021), as we show in Section 3 (Figure 2).

**Asynchronous Federated Learning.** Federated Learning (FL) focuses on training a global model across resource-constrained mobile devices while preserving data privacy (McMahan et al., 2017; Li et al., 2020). In FL, several works also utilize asynchronous training to achieve better time-to-loss performance by addressing device heterogeneity and reducing synchronization delays (Xie et al., 2019; Chen et al., 2020; Nguyen et al., 2022). To enhance scalability, asynchronous hierarchical structures have been proposed, where multiple servers are coordinated by a global server (Wang & Wang, 2022; Mitra & Ulukus, 2023; Xie et al., 2024) or operate in a decentralized manner (Sun et al., 2023; Zuo et al., 2024; Liang et al., 2024). However, these methods require workers to wait for global model broadcasts or for synchronizations of neighboring servers, resulting in significant computational inefficiencies with slow inter-region communication in geo-distributed environments. Moreover, the hierarchical asynchronous FL approaches are primarily designed for low-performance devices, and their performance in large-scale language model training remains underexplored, particularly in integrating momentum—a critical component for effective LLM training—and providing its theoretical convergence analysis.

## 3. Motivation

We aim to minimize a loss function $F(\cdot)$ for a model $\Theta$, collaboratively trained by $N$ workers distributed across multiple geographical regions with heterogeneous operating speeds. The optimization objective is formulated as:

$$\Theta_* = \mathrm{argmin}_\Theta \frac{1}{N} \sum_{i=1}^{N} F_i(\Theta) \qquad (1)$$

where $F_i$ represents the local loss function computed by worker $i$ using its assigned training data. The training data for each worker is assumed to be independently and identically distributed (i.i.d.)[2]. Figure 2 (a) illustrates an example of the geo-distributed training environment with 4 regions and 16 heterogeneous workers.

In this geo-distributed setup, DiLoCo (Section 2) faces significant challenges due to high synchronization costs due to slow inter-region communication bandwidth, high delays and the straggler problem, where the slowest worker bottlenecks the overall training progress. On the other hand, Async-Local-SGD, while designed to mitigate some of these

---

[2]We follow standard practice in assuming i.i.d. data for analytical clarity, but HALoS does not rely on this assumption in practice. See Section 5.4 for further discussion and empirical results.

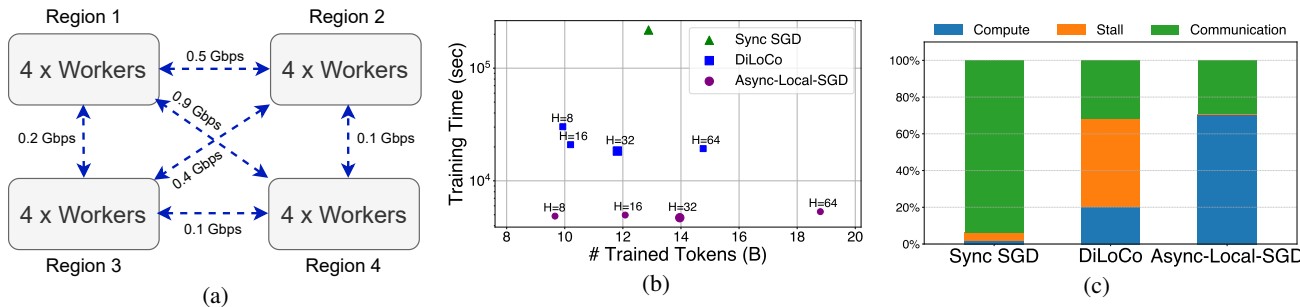

*Figure 2.* (a) A geo-distributed training environment with 4 regions with 16 heterogenous workers where we adapt recent measurement of inter-region communication bandwidths from (Jaghouar et al., 2024). (b) Comparison of training time (in log scale) and token consumption for Pythia-70M model (Biderman et al., 2023) in (a) to reach the same validation loss under various training methods ("H" denotes the number of local updates in workers.) (c) Runtime breakdown analysis of workers for synchronous SGD, DiLoCo (H=32), and Async-Local-SGD (H=32) in (b). We detail the experimental setup in Appendix A.

issues, encounters prolonged wait times when workers communicate with a global parameter server, which is likely to be deployed in a different geographical region.

Although both methods reduce communication overhead by increasing the number of local updates ($H$) performed by workers, this incurs a trade-off: a higher $H$ increases variance in training, requiring more tokens to achieve the same convergence. As shown in Figure 2 (b), while infrequent communication (via larger $H$) accelerates training initially, it ultimately demands more tokens for the same validation loss and makes the convergence slower with too large $H$. Both methods achieve their fastest convergence at $H = 32$.

Figure 2 (c) illustrates the overheads associated with geo-distributed training methods that achieve the fastest times to convergence. For synchronous SGD, communication overhead dominates, consuming 93.7% of worker runtime due to frequent synchronization of large model states across regions. DiLoCo alleviates some cost by enabling multiple local updates, but its strictly synchronous design results in the slowest worker stalling the progress of others, leading to 47.5% of worker runtime being stalled. Lastly, Async-Local-SGD mitigates long stall times but workers still spend 29.2% of their runtime waiting for communication with the global parameter server, either to pull the latest global model or to push the computed local gradients.

## 4. Hierarchical Asynchronous Local SGD

To overcome the fundamental impact of slow communication in geo-distributed training, we propose HALoS, a novel hierarchical distributed optimization framework that is fully asynchronous at each level of its hierarchy. HALoS employs local parameter servers (LPS) within each geographical region, as illustrated in Figure 1. These servers aggregate updates from local workers and accumulate a specified number of updates before communicating with the global parameter server (GPS). This approach reduces

the frequency of communication with the GPS, lowering inter-region communication costs while leveraging fast intra-region links for local updates. By accumulating updates at the LPS level, HALoS effectively relieves the scalability bottleneck at the GPS, allowing for more efficient handling of large-scale distributed training.

### 4.1. Algorithm

Specifically, HALoS operates on three levels (Algorithm 1):

1. Global Parameter Server: The GPS receives accumulated updates $\Delta$ from an LPS (line 2), applies it to the global model $\Theta_i$ using a momentum-based update rule (line 3), then sends the model back to the LPS (line 4).

2. Local Parameter Servers: Each LPS maintains a local model $\theta_t$, receives gradients $\delta$ from workers (line 9), and applies them using a momentum-based update rule (line 11). After $K$ updates (line 13), it sends the accumulated updates $\Delta$ to the GPS (line 14) and continues updating its local model with the workers. Upon receiving an updated global model $\Theta_i$ (line 17), the LPS merges it with its local model using a weighted average controlled by $\alpha$ (line 19). Both worker gradients ($\delta$) and the latest global model ($\Theta_i$) are placed in the same queue, and updates and merging occur in that order.

3. Workers: Each worker performs $H$ local gradient descent steps on its assigned data (line 22) before sending the resulting gradient $\delta$ to its LPS (line 25).

The hyperparameters $K$ and $\alpha$ are central to HALoS's ability to balance communication efficiency and convergence quality. First, $K$ regulates the frequency of LPS-to-GPS communication, effectively reducing the computational load on the GPS while controlling the variance of gradients from geographically diverse LPSs. Second, $\alpha$ determines the extent to which the global model influences the local model during the merging process. This parameter is crucial for accounting for regional learning progress in asynchronous

**Algorithm 1** HALoS Update Rules.

**Require:** Initial model $\Theta_0$, total training iterations $T$, learning rates $\eta_g, \eta_l, \eta_w$, momentum coefficients $\beta_g, \beta_l$, model merging weight $\alpha$, number of updates in local server $K$, and number of local steps $H$.

---

*Global Parameter Server (GPS):*
1: **for** $i \leftarrow 1$ **to** $T$ **do**
2:    Receive $\Delta$ from an LPS.
3:    $\Theta_i \leftarrow \text{ModelUpdate}(\Theta_{i-1}, \Delta, \eta_g, \beta_g)$
4:    Send $\Theta_i$ to the LPS.
5: **end for**

---

*Local Parameter Server (LPS):*
6: Initialize $\theta_0 \leftarrow \Theta_0, t \leftarrow 0, t_{last} \leftarrow 0$.
7: Schedule all workers $\theta_0$.
8: **repeat**
9:    **if** receive $\delta$ from a worker **then**
10:     $t \leftarrow t + 1$
11:     $\theta_t \leftarrow \text{ModelUpdate}(\theta_{t-1}, \delta, \eta_l, \beta_l)$
12:     Schedule the worker $\theta_t$.
13:     **if** $t - t_{last} = K$ **then**
14:      Send $\Delta \leftarrow \theta_t - \theta_{t_{last}}$ to GPS.
15:     **end if**
16:    **end if**
17:    **if** receive $\Theta_i$ from GPS **then**
18:     $t_{last} \leftarrow t$
19:     $\theta_t \leftarrow (1-\alpha)\theta_t + \alpha\Theta_i$
20:    **end if**
21: **until** *training completed*

---

*Worker $\theta_{t,0} \leftarrow \theta_t$ scheduled:*
22: **for** $i \leftarrow 1$ **to** $H$ **do**
23:    $\theta_{t,i} \leftarrow \theta_{t,i-1} - \eta_w \nabla F_{i-1}(\theta_{t,i-1})$
24: **end for**
25: Send $\delta \leftarrow \theta_{t,H} - \theta_{t,0}$ to the LPS

---

communication, maintaining an optimal interplay between local and global training dynamics.

**Comparison with Existing Hierarchical Design.** Existing asynchronous hierarchical structures for federated learning often struggle with efficient, scalable optimization in geo-distributed settings. For instance, methods like FedAH (Wang & Wang, 2022) and Timely-AHFL (Mitra & Ulukus, 2023) impose semi-synchronous constraints, requiring regional local servers to wait for fully transmitted global models, causing high delays. In contrast, our method enables fully asynchronous communication, allowing LPSs and colocated workers to update local models continuously, improving efficiency. We empirically show that HALoS's design outperforms existing methods in geo-distributed LLM training (Section 5.2).

On the other hand, HGA-FL (Xie et al., 2024) also hides LPS-to-GPS communication and aims to utilize worker resources efficiently but requires the GPS to maintain multiple model replicas, increasing memory and computational demands. Our fully asynchronous design, leveraging LPS-side update accumulation and global model merging, eliminates these inefficiencies.

## 4.2. Convergence Analysis

Next, we present the theoretical guarantee for Algorithm 1, accompanied by a detailed proof and a formal notation description in Appendix E. Our analysis begins by introducing several standard assumptions commonly used in asynchronous distributed optimization. The first two assumptions correspond to typical SGD settings. The third is standard in distributed optimization, and the last two are frequently adopted in asynchronous optimization, *e.g.*, (Nguyen et al., 2022).

**Assumption 4.1** (L-smoothness). Local loss functions are L-smooth, *i.e.*, there exists $L > 0$, such that for any worker $i$, $\|\nabla F_i(x) - \nabla F_i(y)\| \leq L\|x - y\|$.

**Assumption 4.2** (Unbiased gradient and bounded variance). The local stochastic gradients are unbiased estimator of full-batch gradients with bounded variance.

**Assumption 4.3** (Bounded heterogeneity). At every iteration, the variance of the aggregated gradients sent from an LPS to the GPS is bounded. Formally, we have $\mathbb{E}\|\nabla_l(\Theta) - \nabla F(\Theta)\|^2 \leq \sigma^2$, where $\nabla_l(\Theta)$ represents the aggregated gradients of the model $\Theta$ on LPS $l$.

**Assumption 4.4** (Bounded gradients). For any worker $i$, the local gradients satisfy $\|\nabla F_i(x)\| \leq G$.

**Assumption 4.5** (Bounded staleness). The norm of the model difference between local training and global updates, caused by asynchronous training, is bounded by $D_g$ for GPS and $D_l$ for LPS, respectively.

**Theorem 4.6.** *Denote $\beta_g$ and $\beta_l$ to be the global and local momentum hyperparameters, and the learning rate satisfies $\eta_0 \geq \eta_t \geq \eta_{\min}$ for all $t$. Under certain assumptions, the following holds:*

$$\min_{1 \leq t \leq T} \mathbb{E}\|\nabla F(\Theta_t)\|^2 \leq \frac{4(F(\Theta_0) - F(\Theta_*))}{\eta_m T}\left(1 + \frac{1}{1 - \beta_g}\right)$$
$$+ \frac{\eta_0}{\eta_m}\frac{1}{\beta_g^3}\left(3 + 12L\eta_0 + \frac{6L\eta_0}{(1-\beta_g)^2}\right)$$
$$\times \left(\frac{G\sigma^2}{(1-\beta_l)(1-\beta_g)} + L^2 D_g^2 + L^2 D_l^2\right)$$
$$= \mathcal{O}\left(\frac{F(\Theta_0) - F(\Theta_*)}{\eta_m(1-\beta_g)T}\right)$$
$$+ \gamma\underbrace{\mathcal{O}\left(\frac{G\sigma^2}{(1-\beta_l)(1-\beta_g)}\right)}_{\text{Hierarchical structure}} + \gamma\underbrace{\mathcal{O}\left(L^2(D_g^2 + D_l^2)\right)}_{\text{Asynchronous update delay}} \quad (2)$$

*where $\gamma = \mathcal{O}\left(\frac{\eta_0^2 L}{\eta_m \beta_g^3(1-\beta_g^2)}\right)$.*

As with previous distributed optimization algorithms (Xie et al., 2019; Wang et al., 2020; Li et al., 2023), our results include a diminishing term $\mathcal{O}(1/T)$ associated with the initial bias and a constant term specific to our hierarchical asynchronous optimization setting. The first component of the

constant term arises from the variance of global and local updates in the hierarchical structure, while the second is due to the delays inherent in the asynchronous setting. Our setting is the first attempt to tackle a hierarchical asynchronous setting with momentum updates, so it is not directly comparable with previous works. By setting $D_g^2 = D_l^2 = 0$, we recover the momentum-based bound from (Liu et al., 2020) in terms of $\beta_l$, with an additional $G$ term arising from asynchronous optimization. Moreover, removing the momentum updates restores the result in (Nguyen et al., 2022). Overall, our derived bound is tight considering both perspectives.

**Influence of momentums $\beta_g$ and $\beta_l$.** To achieve effective convergence in training, HALoS incorporates momentum-based updates for both GPS and LPS, with coefficients $\beta_g$ and $\beta_l$. Our theoretical bounds suggest that these two momentum terms have different magnitudes of influence. Specifically, for local momentum $\beta_l$, the bound follows $\mathcal{O}\left(\frac{1}{1-\beta_l}\right)$, in line with prior momentum-based studies, e.g., (Liu et al., 2020), which allows us to use typical values such as 0.9 (Douillard et al., 2023; Liu et al., 2024).

In contrast, for global momentum $\beta_g$, the bound is more complex as $\mathcal{O}\left(\frac{1}{\beta_g^3(1-\beta_g)^3}\right)$. Two insights emerge:

1. The factor $(1 - \beta_g)$ appears with higher powers than usual momentum bound, preventing $\beta_g$ from being set too high (e.g., 0.9).

2. The additional $\beta_g^3$ factor rules out very small values.

Consequently, $\beta_g$ should be chosen as a trade-off value, such as 0.5, which approximately minimizes $\frac{1}{x^3(1-x)^3}$. Remarkably, our experiments confirm that these choices of $(\beta_g, \beta_l)$ are indeed optimal (Section 5.2).

**Influence of heterogeneity $\sigma^2$ and staleness $D_g$, $D_l$.** Taking both heterogeneity and staleness into account provides additional insight into the choice of momentum. Delays and staleness introduce bias in gradient estimation, and the theory indicates that stale information can inflate effective noise and reduce the benefit of higher-level momentum. When workers in different LPSs operate on highly heterogeneous data, the aggregated global gradient $\nabla_g(\Theta)$ may exhibit conflicting directions. In such cases, GPS-level momentum risks "averaging" non-stationary and contradictory signals, potentially exacerbating the mismatch and slowing convergence. Consequently, it is preferable not to use or use a smaller $\beta_g$. By contrast, momentum at the LPS level works on more homogeneous data and tends to be more stable. We provide empirical evidence of this analysis in Section 5.4.

## 5. Experiments

In this section, we evaluate the performance of HALoS in training LLMs within geo-distributed environments through custom execution trace-driven simulations. Our evaluation covers four aspects: (1) end-to-end pretraining performance compared to baseline methods (Section 5.1); (2) the impact of individual techniques and hyperparameters of HALoS (Section 5.2); (3) generalization across diverse cluster configurations and model families (Section 5.3); and (4) robustness to data heterogeneity (Section 5.4).

**Evaluation Task.** We measure wall-clock times and number of tokens required to train LLMs to achieve specific model performance (i.e., test accuracy or validation loss) in a geo-distributed setup. To ensure reproducible evaluations, we train the Pythia models (Biderman et al., 2023) using the deduplicated Pile dataset (Gao et al., 2020). Pythia provides the order of trained tokens and fine-grained snaphots of models during full training across various LLM sizes. For all evaluations, we use the same learning rates and batch sizes used in training the models and adhere to standard optimization practices in LLM training: learning rates follow a linear warmup phase and decay to 10% of their maximum value using a cosine schedule, automatic mixed precision training with float16 (Micikevicius et al., 2017; PyTorch, 2025), AdamW optimizer with 0.1 weight decay (Loshchilov, 2017), and gradient clipping set to 1.0.

**Experimental Setup.** We simulate a geo-distributed environment consisting of four geographical regions, each hosting four workers, as illustrated in Figure 2 (a). To model realistic wall-clock training times, we utilize recent measurements of inter-region network bandwidths (Jaghouar et al., 2024). The workers are assigned operating speeds uniformly at random within a range of 1 to 10, reflecting the recent rapid advancements in accelerator computation capabilities, where higher values indicate faster workers. We profile the execution of training computations on H100 GPUs and use the profiled time to represent the computation time of the fastest worker. For other workers, their computation times are scaled based on their relative speeds. Our simulator estimates computation time based on the profiled runtime and communication time using widely adopted analytical models (Valiant, 1990; Thakur et al., 2005). Then, it determines the order of updates for each local and global model and executes model updates based on this order. We describe further details about the experimental setup in Appendix A.

**Baselines.** We compare HALoS with two baseline methods: DiLoCo (Douillard et al., 2023) and Async-Local-SGD (Liu et al., 2024). Async-Local-SGD adjusts the number of local steps based on the inverse of worker speeds, reducing gradient staleness by allowing workers to complete computations at similar times. To evaluate the impact of this dynamic adjustment on synchronous methods, we introduce

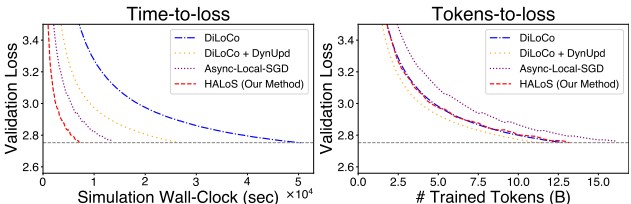

*Figure 3.* Validation loss curves of the Pythia-160M model for different local SGD-based methods to reach the same validation loss: (left) loss vs. simulation wall-clock time and (right) loss vs. number of trained tokens.

"DiLoCo + DynUpd," which integrates dynamic local updates into DiLoCo. Additionally, to ensure a fair comparison with Async-Local-SGD, HALoS incorporates both dynamic local updates and the delayed Nesterov optimizer used in Async-Local-SGD.

**Hyperparameters.** We conduct a hyperparameter sweep for HALoS and each baseline method using the smallest model, Pythia-70M, and use the best-performing hyperparameters for all subsequent evaluations. Details of the candidate hyperparameters and the selected values for each method are provided in Appendix D.

### 5.1. End-to-end Training Performance

In this section, we evaluate the training efficiency of HA-LoS compared to local SGD-based methods and assess the generalization capabilities of LLMs trained with HALoS.

**Compared to Local SGD-based Methods.** We evaluate the end-to-end training performance of HALoS against existing local SGD-based methods. Specifically, we train the Pythia-160M model from scratch on the first 12.9B tokens from the Pile dataset (with 0.6B tokens used for the initial warmup), measure the final validation loss, and then train the same initial model using different methods until each reaches the same validation loss. Following the original Pythia model configuration, we use a global batch size of 1,024 where each of the 16 workers uses a mini-batch size of 64 with a sequence length of 2,048 per local step.

Figure 3 illustrates the validation loss curves as a function of training time (left) and the number of trained tokens (right) for each method. HALoS achieves a 7.1× faster training speed compared to DiLoCo, which suffers from straggler issues and high synchronization delays. When dynamic local steps are applied to DiLoCo (DiLoCo + DynUpd), performance improves by mitigating the straggler problem through adaptive step adjustments based on worker speeds. This modification also reduces token requirements by 10.8% due to increased synchronization frequency. Nevertheless, HALoS outperforms DiLoCo + DynUpd by 3.8×, benefiting from its hierarchical asynchronous design that effectively conceals synchronization delays.

*Table 1.* Training performance of Pythia-70M, Pythia-160M, and Pythia-410M under various training methods. The table shows normalized simulated wall-clock times (relative to HALoS) and the number of tokens used to reach the same validation loss. The corresponding validation loss curves are detailed in Appendix B.

| | PYTHIA-70M | | PYTHIA-160M | | PYTHIA-410M | |
| --- | --- | --- | --- | --- | --- | --- |
| | TIME | TOKENS | TIME | TOKENS | TIME | TOKENS |
| DiLoCo | 7.2 | 11.8B | 7.1 | 12.9B | 7.5 | 12.3B |
| DiLoCo + DynUpd | 4.0 | 10.2B | 3.8 | 11.5B | 3.9 | 10.8B |
| Async Local-SGD | 1.8 | 14.0B | 1.9 | 16.1B | 2.1 | 16.1B |
| HALoS (Ours) | **1.0** | 12.3B | **1.0** | 13.2B | **1.0** | 11.8B |

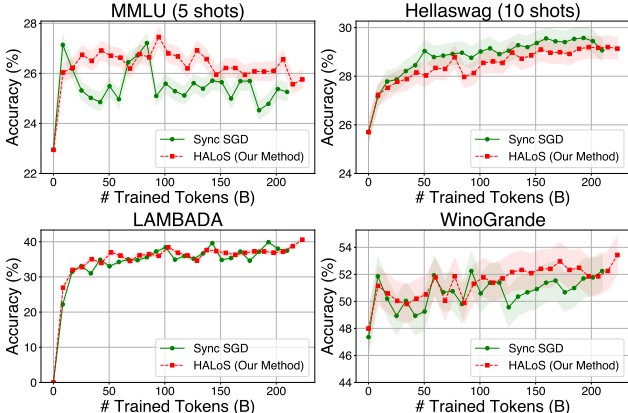

*Figure 4.* Test accuracies of the Pythia-160M model on the MMLU, Hellaswag, LAMBADA, and WinoGrande benchmarks, plotted against the number of trained tokens. The results compare synchronous SGD and HALoS. The shaded regions represent the standard errors in benchmark evaluations. We use 5-shot evaluations for MMLU and 10-shot evaluations for Hellaswag. Results for four additional benchmarks (SciQ, PIQA, ARC-Challenge, and ARC-Easy) are provided in Appendix B.

Compared to Async-Local-SGD, HALoS overcomes limitations related to communication delays. Async-Local-SGD workers are periodically idle due to the need to fetch updated global models and upload gradients, which increases the number of required local steps and adversely affects convergence. In contrast, HALoS delivers 1.9× faster convergence and reduces token consumption by 18.0%.

We extend the analysis across models of varying sizes, as detailed in Table 1. The advantages of HALoS persist across scales. For example, when training the Pythia-410M model, HALoS achieves a 7.5× speedup over DiLoCo, 3.9× over DiLoCo + DynUpd, and 2.1× over Async-Local-SGD.

**Benchmarks Performance.** We further evaluate the test accuracies of HALoS on four representative benchmarks: MMLU (Hendrycks et al., 2020), Hellaswag (Zellers et al., 2019), LAMBADA (Paperno et al., 2016), and Wino-Grande (Sakaguchi et al., 2021). We fully train the Pythia-160M model with HALoS until it matches or exceeds the performance of the publicly available model checkpoints

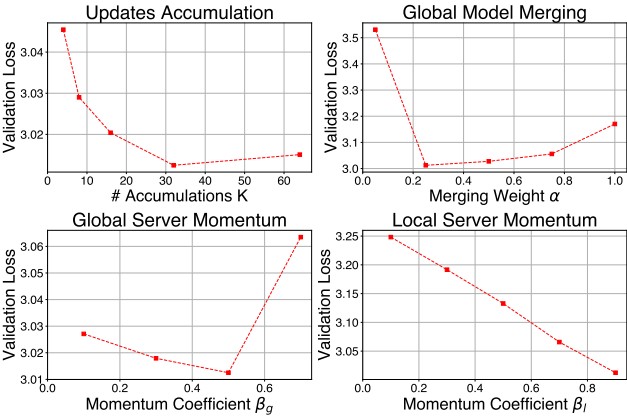

*Figure 5.* Hyperparameter sensitivity of HALoS: validation losses of Pythia-70M after training 12.9B tokens, varying (top left) updates accumulation $K$, (top right) merging weight $\alpha$, (bottom left) global server momentum $\beta_g$, and (bottom right) local server momentum $\beta_l$. The best loss is achieved with $K = 32$, $\alpha = 0.25$, $\beta_g = 0.5$, and $\beta_l = 0.9$.

trained by fully synchronous SGD on one epoch of the Pile dataset (207B tokens). In practice, this requires training on 223B tokens to ensure broad coverage and stable convergence. As shown in Figure 4, the HALoS-based model consistently meets or surpasses the baseline's accuracy. HALoS achieves the performance with a 68.6× speedup in training time by effectively addressing the high communication delays and straggler problem of synchronous SGD in geo-distributed environments.

### 5.2. Ablation Studies

We evaluate the impact of different values of hyperparameters used in HALoS: updates accumulation ($K$), global model merging weight ($\alpha$), and server momentums ($\beta_l$ and $\beta_g$). We train Pythia-70M model on 12.9B tokens and report the final validation losses using different values of hyperparameters.

**Impact of $K$.** As we describe in Section 4.1, $K$ plays a pivotal role in achieving scalable structure by managing the communication frequency between LPSs and GPS. It also reduces the overhead associated with global model updates in GPS. As shown in Figure 5 (top left), HALoS demonstrates robust performance across various $K$ values, with the best performance observed at $K = 32$, where each LPS has 4 workers. Notably, we observe that a high $K$ not only reduces the overhead but also mitigates the variance of gradients from different LPSs. This occurs because the accumulation of a minimum number of updates ensures stability, irrespective of the differences in LPS-to-GPS communication bandwidths.

**Impact of $\alpha$.** The global model merging weight $\alpha$ is a critical parameter that balances the contributions of global

*Table 2.* Training performance of HALoS with ❶ Global and Local Momentums ($\beta_l = 0.9, \beta_g = 0.5, K = 4, \alpha = 1.0$), ❷ Global Model Merging ($\alpha = 0.25$), and ❸ Local Server Updates Accumulation ($K = 32$), compared to FedAH (Wang & Wang, 2022). All methods are trained until they reach the same validation loss. The training times are normalized to HALoS (❶ + ❷ + ❸).

| METHOD | TIME-TO-LOSS | TOKENS-TO-LOSS |
|---|---|---|
| FEDAH | 3.3 | 38.1B |
| HALoS (❶) | 1.6 | 19.3B |
| HALoS (❶ + ❷) | 1.3 | 15.6B |
| HALoS (❶ + ❷ + ❸) | **1.0** | 12.3B |

and local models during the merging process. As shown in Figure 5 (top right), the validation loss is minimized when $\alpha = 0.25$. When $\alpha$ is too low (approaching 0), the local models become overly isolated, failing to incorporate global learning progress effectively. This isolation leads to divergence across regions and hinders convergence, resulting in suboptimal performance. Conversely, when $\alpha$ is too high (close to 1), updates made during communication between LPSs and GPS are not effectively reflected after the merging process, leading to degraded convergence behavior.

**Impact of $\beta_g$ and $\beta_l$.** The training efficiency of HALoS is significantly influenced by the global momentum $\beta_g$ and the local momentum $\beta_l$. As shown in Figure 5 (bottom left and right), the optimal values are $\beta_g = 0.5$ and $\beta_l = 0.9$, which align with our theoretical analysis in Section 4.2. The lower optimal value for $\beta_g$ supports the theoretical insight that global momentum needs to be more conservative to mitigate the accumulation of conflicting signals from diverse regions. In contrast, the higher optimal value for $\beta_l$ leverages the relative homogeneity of updates within local servers, allowing for more aggressive use of momentum.

**Contributions of Key Techniques in HALoS.** Table 2 illustrates the impact of HALoS's key techniques on training efficiency compared to FedAH (Wang & Wang, 2022), which applies staleness scaling with a polynomial form ($\beta = 0.5$) following prior work (Xie et al., 2019; Nguyen et al., 2022). First, integrating local and global momentums (❶) accelerates convergence, achieving the same validation loss as FedAH with 2.0× fewer tokens. Second, applying global model merging (❷) prevents wasted local computations and better balances local and global training progress, further reducing training time and token requirements by 1.2×. Finally, accumulating updates at local servers (❸) improves training efficiency by lowering the variance of accumulated updates across regions, achieving an overall 3.3× faster convergence than FedAH.

### 5.3. Generalization Studies

In this section, we assess HALoS 's generalization by measuring its performance across heterogeneous worker distri-

*Table 3.* Relative training time on heterogeneous clusters (2, 4, 4, 6 workers per region) when training Pythia-70M. HALoS (Naive Grouping) assigns one LPS to each region, whereas HALoS (Consistent Grouping) deploys 1, 2, 2, and 3 LPSs so that each LPS serves exactly two workers. Using the same hyperparameters tuned for the homogeneous (4, 4, 4, 4 workers) setting, HALoS (Naive Grouping) diverges.

| METHOD | TIME-TO-LOSS |
|---|---|
| ASYNC-LOCAL-SGD | 1.7 |
| HALoS (NAIVE GROUPING) | - |
| HALoS (CONSISTENT GROUPING) | **1.0** |

butions, different inter- and intra-region bandwidths, and multiple LLM families under the same hyperparameter setting. We also compare HALoS with fully synchronous model-parallel baselines in geo-distributed environments.

**Heterogeneous Clusters.** We evaluate HALoS under heterogeneous worker distributions (2,4,4,6 workers per region), comparing against our strongest baseline, Async-Local-SGD. As shown in Table 3, when workers were naively grouped into one LPS per region, resulting in significantly varying workers per LPS (2 to 6), training diverges due to large discrepancies in update progresses across LPSs. To address this, we employ a **consistent grouping strategy** ensuring each LPS manages exactly two workers, resulting in 1,2,2,3 LPSs per region. Using this strategy, HALoS efficiently coordinates learning across heterogeneous clusters, achieving 1.7× faster convergence than Async-Local-SGD. Here, we trained the Pythia-70M model as described in Section 5.1. For consistent grouping, hyperparameters remained unchanged, except for adjustments to local updates accumulation ($K$) to 8 and local momentum update delay ($d_l$) to 4, reflecting the smaller number of workers per LPS.

**Different Network Bandwidths.** We evaluate the impact of network bandwidths on HALoS by comparing its performance against local SGD methods in Appendix C.1. Even with 2× faster inter- and intra-region network bandwidths, HALoS consistently outperforms the baselines, achieving 5.7× and 1.5× faster convergence than DiLoCo and Async-Local-SGD, respectively.

**Different Model Families.** We evaluate the generality of HALoS across three popular LLM families, Llama (Dubey et al., 2024), Qwen (Yang et al., 2025), and Pythia (Biderman et al., 2023), using exactly the same hyper-parameters tuned on Pythia (Appendix C.2). HALoS consistently outperforms the strongest baseline, Async-Local-SGD, achieving 2.1× faster convergence on Llama-70M and 2.3× on Qwen-70M without any additional tuning, underscoring its generality to architectural choices of LLMs.

**Comparison with Model Parallelism Techniques.** HALoS integrates model parallelism (MP) seamlessly by treating each worker as a set of accelerators that jointly train a single

*Table 4.* Training performance of HALoS and Async-Local-SGD on the non-i.i.d. Shakespeare dataset (Caldas et al., 2018) when training the 6-layer Llama model. The table reports character-level test accuracy and normalized training time.

| METHOD | ACCURACY | TIME-TO-ACCURACY |
|---|---|---|
| ASYNC-LOCAL-SGD | 49.2% | 1.6 |
| HALoS (W/ GLOBAL MOMENTUM) | 46.9% | 1.0 |
| HALoS (W/O GLOBAL MOMENTUM) | **50.0%** | **1.0** |

model replica. In Appendix C.3, we further compare HALoS on training Pythia-70M against three fully synchronous MP baselines: data parallelism (DP), DP + pipeline parallelism (PP), and a heterogeneity-aware DP + PP variant. HALoS converges 8.26× faster than the strongest baseline (heterogeneity-aware DP + PP), demonstrating the effectiveness of its hierarchical asynchronous design in avoiding the high synchronization costs (e.g., pipeline stalls) that hamper synchronous MP under heterogeneous accelerator speeds.

### 5.4. Robustness to Data Heterogeneity

We evaluate HALoS on the standard non-i.i.d. split of the Shakespeare dataset, where each character's lines are assigned to a distinct worker (Caldas et al., 2018; McMahan et al., 2017; Li et al., 2023). As shown in Table 4, HALoS without global momentum reaches 50.0% test accuracy 1.6× faster than Async-Local-SGD. With global momentum, HALoS maintains the speed-up but reduces accuracy by 3.1 percentage points, which confirms our analysis in Section 4.2 that averaging conflicting directions into momentum can undermine convergence. These results highlight HALoS's robustness to data heterogeneity and its suitability for geo-distributed LLM training under non-i.i.d. workloads.

**Experiment Details:** We train a 6-layer Llama-style model (hidden size 128) for next-character prediction (among 79 unique characters) given the previous 80. The training set contains 3.15 M samples; the same 16 workers in Appendix A process 64-sample batches each. Evaluation uses 0.52 M held-out samples. Training runs for one epoch with a peak local learning rate of 0.01. HALoS employs its default hyperparameters, while Async-Local-SGD adjusts local steps (H) from 32 to 16 to avoid divergence.

## 6. Conclusion

We present HALoS, a novel hierarchical asynchronous optimization framework for geo-distributed LLM training. HALoS integrates server-side update accumulation and global model merging to mitigate communication delays and resource heterogeneity, supported by a tight convergence proof and insights into the effects of momentum, delays, and staleness in the hierarchical asynchronous structure. Empirically, we demonstrate that HALoS achieves significantly faster convergence than the baselines methods.

## Acknowledgements

Kim and Akella are supported by NSF grants CNS-2105890 and CNS-2232135, and by gifts from Meta and Cisco Research. Z. Wang is in part supported by NSF Award 2145346 (CAREER) and 2212176. This research has been supported by computing support on the Vista GPU Cluster through the Center for Generative AI (CGAI) and the Texas Advanced Computing Center (TACC) at the University of Texas at Austin.

## Impact Statement

This paper presents a new distributed optimization method to improve the efficiency of LLM training using geographically distributed resources. The proposed hierarchical asynchronous framework, HALoS, addresses slow inter-region communication delays and hardware heterogeneity to improve training efficiency. The main social and ethical implications of this work relate to enabling more resource-efficient and accessible LLM training at scale. While these advances have broad applicability, we do not foresee any direct societal harm arising from this study.

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

# A. Details on Experimental Setup

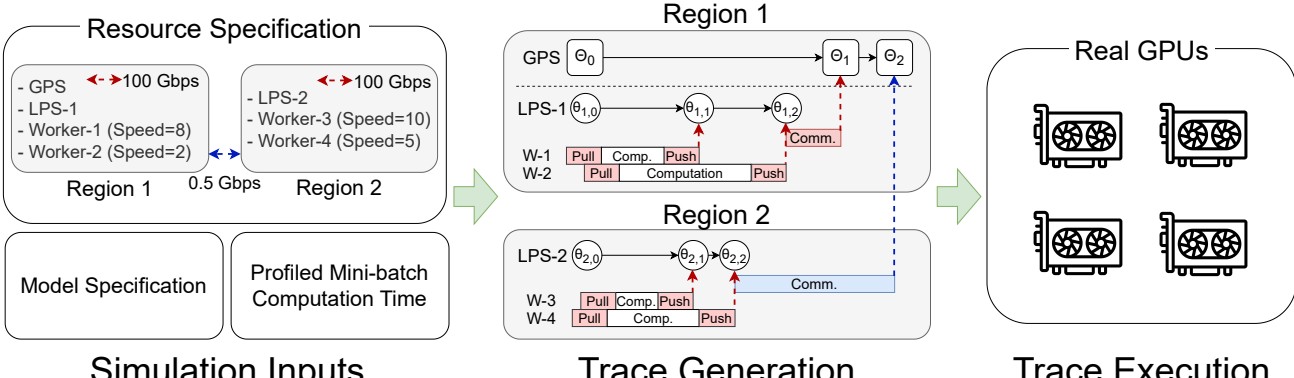

*Figure 6.* **Simulation approach for evaluating geo-distributed training methods.** The simulator accepts: (i) resource specifications (e.g., network bandwidths and the placements of the Global Parameter Server (GPS), Local Parameter Servers (LPSs), and workers along with their relative operating speeds), (ii) model specifications, and (iii) profiled per-step computation time on real GPUs. Using these inputs, it determines the computation and communication events under each training method, calculates the duration of each event, then generates a trace capturing the sequence of model updates (and their dependencies) in GPS and LPSs. Finally, the simulator executes the trace on actual GPUs, respecting the recorded update ordering and dependencies. We release our code publicly at https://github.com/utnslab/halos.

|     | R-1   | R-2   | R-3   | R-4   |
|-----|-------|-------|-------|-------|
| R-1 | 100.0 | 0.537 | 0.935 | 0.202 |
| R-2 | 0.537 | 100.0 | 0.386 | 0.117 |
| R-3 | 0.935 | 0.386 | 100.0 | 0.127 |
| R-4 | 0.202 | 0.117 | 0.127 | 100.0 |

*Table 5.* **Inter- and intra-region communication bandwidths (Gbps).** Four regions are used in the evaluation, following measurements in (Jaghouar et al., 2024) for inter-region bandwidths (0.117 to 0.935 Gbps). The intra-region bandwidth is set to 100.0 Gbps.

| REGIONS | WORKER SPEEDS |     |     |     |
|---------|------|------|------|------|
| R-1     | 10.0 | 9.1  | 3.8  | 2.6  |
| R-2     | 9.4  | 8.0  | 6.3  | 5.8  |
| R-3     | 9.9  | 5.7  | 2.1  | 1.5  |
| R-4     | 9.1  | 8.7  | 5.8  | 1.2  |

*Table 6.* **Relative speeds of the 16 workers.** Speeds are drawn uniformly in the range [1.0, 10.0], reflecting heterogeneous operating speeds of workers.

| MODEL | TIME (MS) |
|-------|-----------|
| PYTHIA-70M  | 238.4  |
| PYTHIA-160M | 623.0  |
| PYTHIA-410M | 1589.7 |

*Table 7.* **Profiled computation times of different models.** Each entry is the duration for one training step (forward, backward, and parameter update) on a single H100 GPU with a mini-batch size of 64, measured on AWS P6 EC2 instances.

This section explains the methodology used to evaluate the performance of various training approaches under geo-distributed environments. A custom trace-driven simulator is developed to capture the effects of network latency, bandwidth constraints, and heterogeneous accelerator speeds. Figure 6 illustrates the overall simulation pipeline.

The simulator takes as input the resource specifications for each region, including both inter- and intra-region bandwidths, as well as the placement of the Global Parameter Server (GPS), Local Parameter Servers (LPSs), and workers, whose speeds vary from 1.0 to 10.0. By default, we place the GPS in the first region (R-1) and use the network bandwidths in Table 5. Table 6 summarizes the relative speeds of 16 workers. Additionally, the simulator takes as input a model configuration along with a profiled computation time measured on real GPUs for one training step. It then uses this profiled time as the simulated computation time for the fastest worker. Table 7 shows these measured times.

With the given inputs and a specified training method, the simulator generates computation and communication events over time, along with the dependencies between model versions, and calculates the duration of each event. For point-to-point communications, such as those used in pulling local models by workers, the simulator uses the well-known $\alpha$–$\beta$ model (Valiant, 1990), given by

$$T_{\text{p2p-comm}} = \alpha + \beta\,C,$$

where $\alpha$ denotes the propagation delay, $\beta$ is the inverse of network bandwidth, and $C$ is the data size. For collective communications, such as the all-reduce operation in DiLoCo, the simulator assumes a widely used ring-based implementation. Specifically, it searches for the ring that maximizes the communication bandwidth of the slowest link and calculates the

communication time using the standard analytical model (Thakur et al., 2005):

$$T_{\text{all-reduce}} = 2 \cdot \frac{(N-1)\,C}{N\,B},$$

where $N$ is the number of participating workers, $B$ is the slowest bandwidth, and $C$ is the data size.

To account for heterogeneous worker speeds, the simulator calculates per-worker computation time by scaling the profiled computation time. Given $T_{\text{profiled}}$ as the time measured for the fastest worker with speed $S_{\text{fastest}}$, the time to perform $H$ local steps on a worker with speed $S$ is computed as:

$$T_{\text{compute}} = H \times T_{\text{profiled}} \times \frac{S_{\text{fastest}}}{S}.$$

Slower workers thus incur proportionally longer computation times, while faster ones complete local steps more quickly.

Once all communication and computation times have been determined, the simulator generates a final trace of model update events at each LPS and the GPS. The order of updates is strictly enforced so that the correct model version is referenced at every update. By replaying this trace on real GPUs, the simulator provides an accurate depiction of how different distributed optimization algorithms perform when subject to realistic network latencies and resource heterogeneity.

# B. Extended Experimental Results

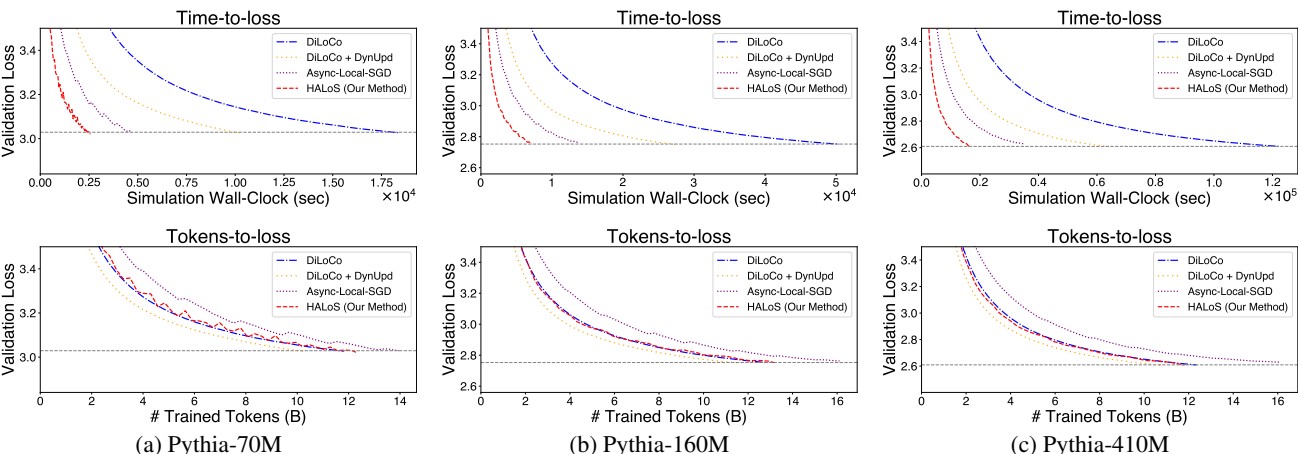

*Figure 7.* Validation loss curves for Pythia-70M, Pythia-160M, and Pythia-410M models trained using different methods from Table 1.

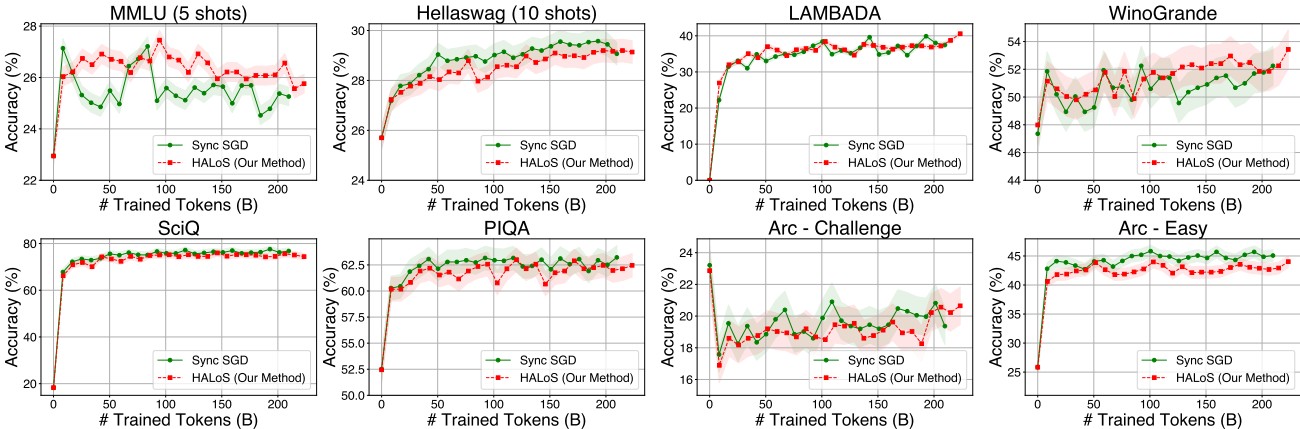

*Figure 8.* Extended benchmark performance results corresponding to Figure 4 for Pythia-160M trained with fully synchronous SGD and HALoS. At the bottom, we additionally includes results for the SciQ, PIQA, Arc-Challenge, and Arc-Easy benchmarks.

We present the validation loss curves for all evaluated models from Table 1 in Figure 7. Additionally, extended benchmark performance results, including four additional benchmarks (SciQ (Welbl et al., 2017), PIQA (Bisk et al., 2020), Arc-Challenge, and Arc-Easy (Clark et al., 2018)), are shown in Figure 8.

# C. Additional Experimental Results

## C.1. Impact of Network Bandwidths

| | 1× Network Bandwidths | | 2× Network Bandwidths | |
| | Time | Tokens | Time | Tokens |
|---|---|---|---|---|
| DiLoCo | 7.2 | 11.8B | 5.7 | 11.8B |
| DiLoCo + DynUpd | 4.0 | 10.2B | 2.3 | 10.2B |
| Async Local-SGD | 1.8 | 14.0B | 1.5 | 14.8B |
| HALoS (Ours) | **1.0** | 12.3B | **1.0** | 13.2B |

*Table 8.* Comparison of training performance for the Pythia-70M model using different methods under the default (1×) and 2× network bandwidths.

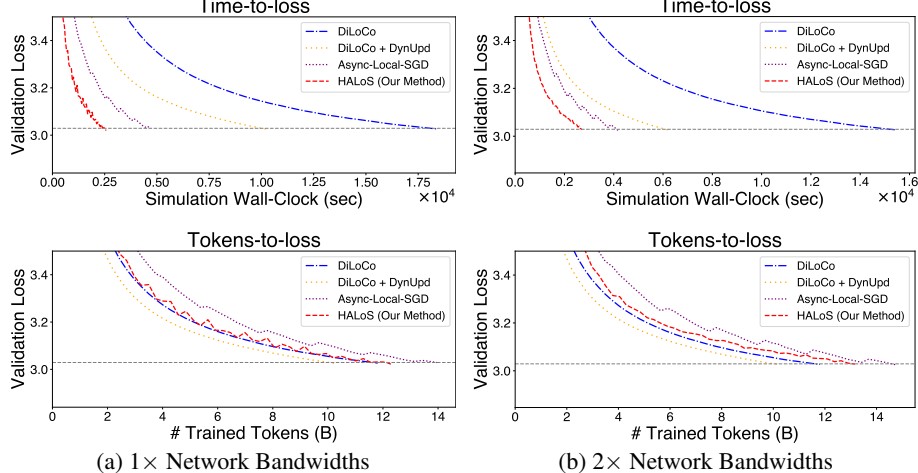

(a) 1× Network Bandwidths      (b) 2× Network Bandwidths

*Figure 9.* Validation loss curves for training the Pythia-70M model with different methods under (a) the default (1×) network bandwidths and (b) 2× network bandwidths. These results correspond to those presented in Table 8.

We evaluate the impact of varying network bandwidths on the training performance of HALoS and three baseline methods: DiLoCo, DiLoCo + DynUpd, and Async-Local-SGD. Specifically, we train the Pythia-70M model using the same configuration detailed in Section 5.1, comparing the default (1×) network bandwidths with 2× faster inter- and intra-machine bandwidths. As shown in Table 8, even when the network bandwidth is doubled, HALoS continues to outperform the baselines, providing up to 5.7×, 2.3×, and 1.5× faster convergence compared to DiLoCo, DiLoCo + DynUpd, and Async-Local-SGD, respectively. Figure 9 illustrates the validation loss curves under both bandwidth settings.

## C.2. Generalization to Different LLM Families

| Model | Training Speedup of HALoS |
|---|---|
| Llama-70M | 2.1× |
| Qwen-70M | 2.3× |
| Pythia-70M | 1.8× |

*Table 9.* Training speedups of HALoS compared to Async-Local-SGD for different LLM families.

To demonstrate the robustness and practical applicability of HALoS across diverse model architectures, we extended our evaluation beyond the Pythia models to include two additional widely-used LLM families—Llama (Dubey et al., 2024) and Qwen (Yang et al., 2025).

As shown in Table 9, HALoS consistently outperforms the strongest baseline (Async-Local-SGD) across all tested LLM architectures. Importantly, the hyperparameters initially optimized for Pythia-70M—without any additional tuning—exhibited strong generalization, achieving even greater relative improvements (up to 2.3× with the Qwen-70M model).

We assessed whether the optimal hyperparameters, identified from our theoretical analysis and validated on the Pythia-70M model, could effectively generalize to these models. To ensure a fair comparison, we selected Llama and Qwen models closely matching Pythia-70M in size, using the same number of layers and hidden dimensions, and conducted the same experiments described in Section 5.1.

### C.3. Comparison with Synchronous Model Parallelism Techniques

| METHOD | RELATIVE TIME-TO-LOSS |
|---|---|
| DP | 85.12 |
| DP+PP | 8.34 |
| HETERO-AWARE DP+PP | 8.26 |
| HALoS | 1.00 |

*Table 10.* Relative training time to reach the same validation loss for various synchronous model parallelism techniques and HALoS.

In HALoS, each worker is a data-parallel group of accelerators, each maintaining a full model replica, enabling seamless integration of Model Parallelism (MP) techniques, including Tensor Parallelism (TP) and Pipeline Parallelism (PP) (Narayanan et al., 2021). Below, we further compare HALoS with strictly synchronous MP methods.

We evaluate relative convergence times across three methods: synchronous DP (DP), synchronous DP with PP (DP+PP), and a heterogeneity-aware version of DP+PP (Hetero-Aware DP+PP). As shown in Table 10, HALoS achieves 8.26× faster convergence compared to the strongest baseline, Hetero-Aware DP+PP. PP allows relatively small activations transferred cross-region and improves convergence speed compared to pure DP (85.12× → 8.34×). However, PP inherently suffers from computational inefficiency from imbalanced pipeline stages (i.e., pipeline bubbles) due to heterogeneous accelerator speeds—even with heterogeneity-aware workload partitioning. In contrast, HALoS effectively mitigates slow inter-region communications and heterogeneous accelerator speeds, achieving superior performance.

We trained Pythia-70M as described in Section 5.1. DP+PP method used a DP degree of 4 and a PP degree of 4, placing pipeline stages across distinct regions. Hetero-Aware DP+PP employed heterogeneity-aware partitioning and a simulated annealing-based heuristic for placement from (Li et al., 2022).

## D. Hyperparameters

In this section, we detail the procedure for deciding the best performing hyperparameters for HALoS and the baseline methods. As described in Section 5, we conduct hyperparameter sweeps using the smallest model, Pythia-70M. Specifically, for each method, the model is pretrained for $6 \times 1024$ steps, using a batch size of 1,024 and a sequence length of 2,048. This corresponds to processing 12.9B tokens, including an initial 300-step warmup phase. In HALoS and Async-Local-SGD, each of the 16 workers uses $1,024/16 = 64$ sequences for each local step. The hyperparameter set yielding the lowest validation loss is selected as the optimal configuration.

Table 11 presents the candidate hyperparameters and highlights the best-performing ones. For HALoS, we employ the delayed Nesterov momentum (Liu et al., 2024) to ensure a fair comparison with Async-Local-SGD. The hyperparameter search for delayed Nesterov updates is conducted using identical candidate sets. For DiLoCo, we observe that the optimal hyperparameters align with those reported in the original paper (Douillard et al., 2023), specifically a learning rate of 0.7 and a momentum coefficient of 0.9.

*Table 11.* The best performing hyperparameters for different training methods. †Delayed Nesterov momentum optimization (Liu et al., 2024) effectively divides the learning rate by the delay interval. We search for learning rates on the scaled ones, as this approach shows more stable comparisons across different momentum update delays.

| | HALoS | Async-Local-SGD | DiLoCo |
|---|---|---|---|
| Global Learning Rate† ($\eta_g/d_g$) | 0.03, 0.05, 0.1, **0.15**, 0.2, 0.25 | 0.03, **0.05**, 0.1, 0.15, 0.2, 0.25 | 0.1, 0.3, 0.5, **0.7**, 0.9 |
| Global Nesterov Momentum ($\beta_g$) | 0.1, 0.3, **0.5**, 0.7, 0.9 | 0.1, 0.3, 0.5, 0.7, **0.9** | 0.1, 0.3, 0.5, 0.7, **0.9** |
| Global Momentum Update Delay ($d_g$) | **2**, 4, 8, 16, 32, 64 | 2, 4, 8, 16, **32**, 64 | **1** |
| Local Learning Rate† ($\eta_l/d_l$) | 0.03, 0.05, 0.1, 0.15, **0.2**, 0.25 | - | - |
| Local Nesterov Momentum ($\beta_l$) | 0.1, 0.3, 0.5, 0.7, **0.9** | - | - |
| Local Momentum Update Delay ($d_l$) | 2, 4, 8, **16**, 32, 64 | - | - |
| Local Updates Accumulation ($K$) | 4, 8, 16, **32**, 64 | - | - |
| Global Model Merging Weight ($\alpha$) | 0.0, **0.25**, 0.5, 0.75, 1.0 | - | - |
| Number of Local Updates ($H$) | **8**, 16, 32, 64 | 8, 16, **32**, 64 | 8, 16, **32**, 64 |

# E. Proof of Theorem 4.6

For clarity, we restate the problem and assumptions. We solve the following distributed optimization problem:

$$\Theta_* = \arg\min_{\Theta} \frac{1}{N} \sum_{i=1}^{N} F_i(\Theta),$$

using Algorithm 1. We have the following assumptions:

**Assumption E.1** (L-smoothness). Local loss functions are L-smooth, *i.e.*, there exists $L > 0$, such that for any worker $i$, $\|\nabla F_i(x) - \nabla F_i(y)\| \leq L\|x - y\|$.

**Assumption E.2** (Unbiased gradient and bounded variance). The local stochastic gradients are unbiased estimator of full-batch gradients with bounded variance.

**Assumption E.3** (Bounded heterogeneity). At every iteration, the variance of the aggregated gradients sent from an LPS to the GPS is bounded. Formally, we have $\mathbb{E}\|\nabla_l(\Theta) - \nabla F(\Theta)\|^2 \leq \sigma^2$, where $\nabla_l(\Theta)$ represents the aggregated gradients of the model $\Theta$ on LPS $l$.

**Assumption E.4** (Bounded gradients). For any worker $i$, the local gradients satisfy $\|\nabla F_i(x)\| \leq G$.

**Assumption E.5** (Bounded staleness). The norm of the model difference between local training and global updates, caused by asynchronous training, is bounded by $D_g$ for GPS and $D_l$ for LPS, respectively.

In the above assumptions, we consider both bounded gradients and bounded staleness. These conditions can be ensured in practice by employing standard stable training techniques, such as gradient clipping. We first clarify the momentum-based update rule of both LPS and GPS. Denote $\nabla_l(\theta_{l,t})$ to be the aggregated gradient in LPS $l$ at time $t$, and $\nabla_g(\Theta_t)$ to be the aggregated gradient in the GPS at time $t$. We denote two operators $lg(l,t)$ and $gl(t)$ for timestamp conversion between GPS and LPS. For LPS, we have the following update rule:

$$m_{l,t+1} = \beta_l m_{l,t} + \nabla_l(\theta_{l,t}),$$
$$\theta_{l,t+1} = (1-\alpha)\theta_{l,t} + \alpha\Theta_{lg(l,t)} - \eta_t\left[(1-\beta_l)\nabla_l(\theta_{l,t}) + \beta_l m_{l,t+1}\right].$$

And for GPS, we have:

$$\nabla_g(\Theta_t) = \frac{1}{M} \sum_{l=1}^{M} \left[(1-\beta_l)\nabla_l(\theta_{gl(t)}) + \beta_l m_{l,gl(t)+1}\right]$$
$$m_{g,t+1} = \beta_g m_{g,t} + \nabla_g(\Theta_t),$$
$$\Theta_{t+1} = \Theta_t - \eta_t\left[(1-\beta_g)\nabla_g(\Theta_t) + \beta_g m_{g,t+1}\right].$$

By the L-smoothness assumption, we have

$$\mathbb{E}[F(\Theta_{t+1}) - F(\Theta_t)] \leq -\eta_t \mathbb{E}[\nabla F(\Theta_t)^T (\nabla_g(\Theta_t) + \beta_g^2 m_{g,t})] + \frac{L\eta_t^2}{2} \mathbb{E}\|\nabla_g(\Theta_t) + \beta_g^2 m_{g,t}\|^2. \tag{3}$$

We bound the two terms in the right side.

To bound the second term, we bound the variance of aggregated gradients on GPS, and the variance of global momentum respectively. For gradients, we have

$$\mathbb{E}\|\nabla_g(\Theta_t) - \nabla F(\Theta_t)\|^2$$

$$= \mathbb{E}\left\|\frac{1}{M}\sum_{l=1}^{M}\left[(1-\beta_l)\nabla_l(\theta_{l,gl(t)}) + \beta_l m_{l,gl(t)} - \nabla F(\Theta_t)\right]\right\|^2$$

$$\leq \frac{2(1-\beta_l)^2}{M}\mathbb{E}\|\nabla_l\theta_{l,gl(t)} - \nabla F(\Theta_t)\|^2 + \frac{2\beta_l^2}{M}\mathbb{E}\|m_{l,gl(t)} - \nabla F(\Theta_t)\|^2$$

$$\leq \frac{2}{M}(\sigma^2 + L^2 D_g^2 + L^2 D_l^2) + \frac{2}{M}\mathbb{E}\left\|\sum_{i=0}^{t}\beta_l^i \nabla_l(\theta_{l,gl(t)-1-i}) - \nabla F(\Theta_t)\right\|^2$$

$$\leq (\sigma^2 + L^2 D_g^2 + L^2 D_l^2) + \frac{\sigma^2}{1-\beta_l} + \frac{1}{1-\beta_l}G.$$

And for the momentum, we have

$$\mathbb{E}\left\|m_{g,t} - \frac{\nabla_g(\Theta_t)}{1-\beta_g}\right\|^2$$

$$= \mathbb{E}\left\|\sum_{i=0}^{t}\beta_g^i(\nabla_g(\Theta_{t-i-1}) - \nabla_g(\Theta_t)) - \sum_{t'=t+1}^{\infty}\beta_g^{t'}\nabla_g(\Theta_t)\right\|^2$$

$$\leq \frac{1}{1-\beta_g^2}\frac{2G}{1-\beta_l} + \frac{G}{1-\beta_l} \leq \frac{(3-\beta_g^2)G}{(1-\beta_g^2)(1-\beta_l)} \leq \frac{3G}{(1-\beta_g^2)(1-\beta_l)},$$

where the last inequality is because for any $t_1$ and $t_2$,

$$\|\nabla_g(\Theta_{t_1}) - \nabla_g(\Theta_{t_2})\|^2 = \left\|\frac{1}{M}\sum_{l=1}^{M}\left[(1-\beta_l)(\nabla_l(\theta_{l,gl(t)}) - \nabla_l(\theta_{l,gl(t_2)})) + \beta_l\left(m_{l,gl(t_1)} - m_{l,gl(t_2)}\right)\right]\right\|^2 \tag{4}$$

$$\leq 2G + \beta_l\sum_{i=0}^{t}\beta_l^i G \leq \frac{2G}{1-\beta_l}. \tag{5}$$

Now we can bound the second term in (3). Denote

$$m_{g,t} = \frac{1}{1-\beta_g}\nabla_g(\Theta_t) + R_{g,t}.$$

We have

$$\nabla_g(\Theta_t) + \beta_g^2 m_{g,t} = \left(1 + \frac{\beta_g^2}{1-\beta_g}\right)(\nabla F(\Theta_t) + (\nabla_g(\Theta_t) - \nabla F(\Theta_t))) + \beta_g^2 R_{g,t},$$

and

$$\mathbb{E}\|\nabla_g(\Theta_t) + \beta_g^2 m_{g,t}\|^2 \leq 3\left(1 + \frac{\beta_g^2}{1-\beta_g}\right)^2 \mathbb{E}\|\nabla F(\Theta_t)\|^2 +$$

$$\left(3\left(1 + \frac{\beta_g^2}{1-\beta_g}\right)^2 + 3\beta_g^4\right)\left(\frac{G\sigma^2}{(1-\beta_l)(1-\beta_g^2)} + L^2 D_g^2 + L^2 D_l^2\right).$$

Denote $\gamma_g = 1 + \frac{\beta_g^2}{1 - \beta_g}$. For the first term in (3), we have

$$
\begin{aligned}
&\nabla F(\Theta_t)^T (\nabla_g(\Theta_t) + \beta_g^2 m_g) \\
&= \gamma_g \|\nabla F(\Theta_t)\|^2 + \nabla F(\Theta_t)^T (\gamma_g \Delta_t + \beta_g^2 R_{g,t}) \\
&\geq \frac{1}{2}(\gamma_g - \beta_g^2)\|\nabla F(\Theta_t)\|^2 - \frac{1}{2}\left(\gamma_g\|\Delta_t\|^2 + \beta_g^2 \|R_{g,t}\|^2\right) \\
&\geq \frac{1}{2}(\gamma_g - \beta_g^2)\|\nabla F(\Theta_t)\|^2 - \frac{1}{2}(\gamma_g^2 + \beta_g^2)\left(\frac{G\sigma^2}{(1-\beta_l)(1-\beta_g^2)} + L^2 D_g^2 + D_l^2\right).
\end{aligned}
$$

Substituting the two terms back in (3) gives:

$$
\begin{aligned}
\mathbb{E}[F(\Theta_{t+1}) - F(\Theta_t)] &\leq -\frac{\eta_t}{2}(\gamma_g - \beta_g^2)\mathbb{E}\|\nabla F(\Theta_t)\|^2 + \frac{\eta_t}{2}(\gamma_g + \beta_g^2)\left(\frac{G\sigma^2}{(1-\beta_l)(1-\beta_g^2)} + L^2 D_l^2 + L^2 D_g^2\right) \\
&\quad + \frac{L}{2}\eta_t^2 \left[3\gamma_g^2 \mathbb{E}\|\nabla F(\Theta_t)\|^2 + (3\gamma_g^2 + 3\beta_g^4)\left(\frac{G\sigma^2}{(1-\beta_l)(1-\beta_g^2)} + L^2 D_g^2 + L^2 D_l^2\right)\right] \\
&= \left(\frac{3}{2}L\eta_t^2 \gamma_g^2 - \frac{\eta_t}{2}(\gamma_g - \beta_g^2)\right)\mathbb{E}\|\nabla F(\Theta_t)\|^2 \\
&\quad + \left(\frac{\eta_t}{2}(\gamma_g + \beta_g^2) + \frac{L}{2}\eta_t^2(3\gamma_g^2 + 3\beta_g^4)\right)\left(\frac{G\sigma^2}{(1-\beta_l)(1-\beta_g^2)} + L^2 D_g^2 + L^2 D_l^2\right).
\end{aligned}
$$

Setting the range of $\eta_t$ satisfies $\eta_m \leq \eta_t \leq \eta_0 \leq \frac{\gamma_g - \beta_g^2}{6L\gamma_g^2}$, the coefficient satisfies:

$$
\frac{3}{2}L\eta_t^2 \gamma_g^2 - \frac{\eta_t}{2}(\gamma_g - \beta_g^2) \leq -\frac{\eta_t}{4}(\gamma_g - \beta_g^2) \leq -\frac{\eta_m}{4}(\gamma_g - \beta_g^2).
$$

Rearranging the above inequality, and taking summation for $0 \leq t \leq T-1$ we have:

$$
\begin{aligned}
\frac{\eta_m}{4}(\gamma_g - \beta_g^2)\mathbb{E}\frac{1}{T}\sum_{t=0}^{T-1}\|\nabla F(\Theta_t)\|^2 &\leq \frac{1}{T}(F(\Theta_0) - F(\Theta^*)) \\
&\quad + \left(\frac{\eta_0}{2}(\gamma_g + \beta_g^2) + \frac{L}{2}\eta_0^2(3\gamma_g^2 + 3\beta_g^4)\right)\left(\frac{G\sigma^2}{(1-\beta_l)(1-\beta_g^2)} + L^2 D_g^2 + L^2 D_l^2\right)
\end{aligned}
$$

With

$$
\gamma_g - \beta_g^2 = \frac{1 - \beta_g + \beta_g^3}{1 - \beta_g},
$$

$$
\gamma_g + \beta_g^2 \leq 2 + \frac{\beta_g^2}{1 - \beta_g},
$$

$$
\gamma_g^2 + \beta_g^4 \leq 2 + \frac{2}{(1 - \beta_g)^2},
$$

and moving the left coefficients to the right side, we simplify the second term:

$$
\begin{aligned}
&\frac{1}{\gamma_g - \beta_g^2}\left(\frac{\eta_0}{2}(\gamma_g + \beta_g^2) + \frac{L}{2}\eta_t^2(3\gamma_g^2 + 3\beta_g^4)\right) \\
&\leq \frac{1 - \beta_g}{\beta_g^3}\left(2 + \frac{\beta_g^2}{1 - \beta_g} + 6L\eta_0\left(2 + \frac{1}{(1 - \beta_g)^2}\right)\right) \\
&\leq \frac{1}{\beta_g^3}\left(3 + 12L\eta_0 + \frac{6L\eta_0}{(1 - \beta_g)^2}\right).
\end{aligned}
$$

This finalize the proof.

