# OpenReview forum: "HALoS: Hierarchical Asynchronous Local SGD over Slow Networks for Geo-Distributed Large Language Model Training"
_ICML.cc/2025/Conference — ICML 2025 poster_

### Official Review · Reviewer_T99B · 2025-03-12

**Overall Recommendation:** 3

**Summary:**

This paper presents a framework for geo-distributed LLM training named HALoS. To reduce the staleness effect, HALoS introduces local parameter server (LPS) for workers within each region, and periodically syncs LPS with global parameter server (GPS). This paper also introduces a convergence analysis for this approach under certain assumptions. The results show that HALoS significantly reduces the amount of sync bottlenecks while maintaining competitive model performance.

## update after rebuttal

I appreciate the reply from the authors during the rebuttal process and have carefully read them. I will upgrade my score because most of the concerns are addressed.

**Claims And Evidence:**

The claims are generally well-supported by the evidences.

**Essential References Not Discussed:**

Not applicable.

**Experimental Designs Or Analyses:**

Q3. For the impact of K (Fig.5), why is the validation loss the highest for smaller K? I assume that a smaller K conrresponds to more frequent sync between LPS and GPS, and a smaller K should be closer to Sync SGD baseline in term of training performance.

Q4. Can you elaborate a bit more on the claim _"the accumulation of a minimum number of updates ensures stability"_ in Section 5.2?

Q5: Fig.7 and Fig. 9shows that DiLoCo+DynUpd has the best convergence speed. Is this because Sync SGD still outperforms Async variants?

**Methods And Evaluation Criteria:**

The methods are mostly sound. But I have a few questions:

Q1: In the geo-distributed training setting, we may not have the same number of workers for each region. The evaluation were conducted on 4x4 setting. How would you split the batch size among regions if the number of workers are difference from region to region?

Q2: when pushing from LPS to GPS, are there mechanism to prevent a staled model weight from "poluting" global model weight, or they just merge with GPS model weight?

**Other Comments Or Suggestions:**

Please refer to the previous comments.

**Other Strengths And Weaknesses:**

Strengths:
* The paper presents an important problem -- straggler mitigation for async sgd, and introduces a hierachical architecture that effectively hide the communication for syncing with computation.
* The speedup is significant compared to baselines.
* This is well-written and easy to follow.

Weakness:
* More models (eg. Llama, QWen) and more model parallelism techniques beyond DP should be explored
* Heterogenous cluster setup (each region having difference number of workers) should also be explored, especially for geo-distributed setting.

**Questions For Authors:**

Please refer to the previous comments.

**Relation To Broader Scientific Literature:**

Outside the geo-distributed training area, this paper is also related to model parallel training (under strictly synchronous setting).

**Theoretical Claims:**

I checked the correctness of the Theorems in Section 4.2, but I am not entirely sure if the assumptions made are valid or realistic.

---

> ### Author Rebuttal · Authors · 2025-04-01
>
> # Clarification on Questions
> **(Q1)** We assign the same mini-batch size per worker. For heterogeneous clusters, we avoid ineffective convergence from imbalanced numbers of workers per LPS by adopting **a consistent grouping strategy** that assigns an equal number of workers to each LPS. For example, if four regions originally have (2,4,4,6) workers, we deploy (1,2,2,3) LPSs, each covering two workers. Empirical results and further discussion are provided in the last section ([W3]) of this response.
>
> **(Q2)** We apply momentum at both local and global levels to mitigate gradient/model staleness effectively (Figure 5). We observe that this approach surpasses prior methods (e.g., gradient penalty), which aligns with previous findings (see Section 2). Our theoretical analysis (Section 4.2) and empirical results (Section 5.2) demonstrate **optimal momentum configurations that minimize staleness effects while maintaining strong performance.** Exploring more advanced, staleness-aware momentum update techniques is left as our future work.
>
> **(Q3,Q4)** We clarify that each LPS accumulates updates during LPS-GPS communication and then $K$ additional updates (Algorithm 1). When $K$ is too small (e.g., $K=1$), regions with higher LPS-to-GPS bandwidth (up to 0.9 Gbps in Figure 1(a)) can push updates more frequently than those with slower links (as low as 0.1 Gbps), causing high variance. By using a modest $K$, each LPS accumulates enough updates before syncing with GPS, ensuring more balanced contributions and reducing variance (Section 5.2).
>
> **(Q5)** We clarify that HALoS consistently achieves **the fastest wall-clock time** to reach the same target loss (top row in Fig. 7 and 9). For example, when training Pythia-410M (Table 1, Fig. 7), **HALoS achieves 3.9x faster convergence** than DiLoCo+DynUpd, with less than 10% additional tokens.
>
> ---
> # [W1] Different Models
> We evaluate HALoS using different LLM families (Llama and Qwen) and observe that HALoS consistently outperforms the baseline. Please see the 'Empirical Validation on Different LLM Families' section in our response to Reviewer Lzk8.
>
> ---
> # [W2] Model Parallelism
> We thank the reviewer for highlighting model parallelism (MP). In HALoS, each worker is a data-parallel group of accelerators, each maintaining a full model replica, enabling seamless integration of MP techniques (e.g., tensor and pipeline parallelism). Below, we further compare HALoS with strictly synchronous MP methods.
>
> **Comparison with Strictly Synchronous MP:** We evaluate relative convergence times across three methods: synchronous DP (DP), synchronous DP with PP (DP+PP), and a heterogeneity-aware version of DP+PP (Hetero-Aware DP+PP).
>
> |Method|Time-to-Loss (Relative to HALoS)|
> |-|-|
> |DP|85.12|
> |DP+PP|8.34|
> |Hetero-Aware DP+PP|8.26|
> |HALoS (Ours)|1.00|
>
> As shown in the above table, **HALoS achieves 8.26x faster convergence** compared to the strongest baseline, Hetero-Aware DP+PP. PP allows relatively small activations transferred cross-region and improves convergence speed compared to pure DP (85.12 → 8.34). However, PP inherently suffers from computational inefficiency from imbalanced pipeline stages (i.e., pipeline bubbles) due to heterogeneous accelerator speeds—even with heterogeneity-aware workload partitioning. In contrast, HALoS effectively mitigates slow inter-region communications and heterogeneous accelerator speeds, achieving superior performance.
>
> **Experiment Details:** We trained Pythia-70M as in Section 5.1. DP+PP method used a DP degree of 4 and a PP degree of 4, placing pipeline stages across distinct regions. Hetero-Aware DP+PP employed heterogeneity-aware partitioning and a simulated annealing-based heuristic for placement from [1].
>
> [1] Dacheng Li et al. AMP: Automatically Finding Model Parallel Strategies with Heterogeneity Awareness. NeurIPS 2022.
>
> ---
> # [W3] Heterogeneous Cluster Setup
> We evaluate HALoS under heterogeneous worker distributions (2,4,4,6 workers per region), comparing against our strongest baseline, Async-Local-SGD.
>
> |Method|Relative Time-to-Loss|
> |-|-|
> |Async-Local-SGD|1.7|
> |HALoS (Naive grouping: 4 LPSs with 2, 4, 4, 6 workers)|-|
> |HALoS (Consistent grouping: 8 LPSs with 2 workers each)|1.0|
>
> When workers were naively grouped into one LPS per region, resulting in significantly varying workers per LPS (2 to 6), convergence slowed due to large discrepancies in update progresses across LPSs. To address this, we employ a consistent grouping strategy ensuring each LPS manages exactly two workers, resulting in 1,2,2,3 LPSs per region. Using this strategy, HALoS efficiently coordinates learning across heterogeneous clusters, **achieving 1.7x faster convergence** than Async-Local-SGD.
>
> **Experiment Details:** We trained Pythia-70M as in Section 5.1. For consistent grouping, hyperparameters remained unchanged, except for adjustments to local updates accumulation ($K=8$) and local momentum update delay ($d_l=4$), reflecting the smaller number of workers per LPS.

---

### Official Review · Reviewer_Lzk8 · 2025-03-14

**Overall Recommendation:** 4

**Summary:**

This paper presents HALoS, a hierarchical asynchronous optimization framework for training large language models (LLMs) across geographically distributed hardware. HALoS addresses communication bottlenecks by using local parameter servers (LPSs) within each region and a global parameter server (GPS) that merges updates across regions. This design minimizes expensive inter-region communication while leveraging fast intra-region links. The framework allows asynchronous updates at each level, enabling efficient communication and computation overlap. The paper provides theoretical convergence guarantees for HALoS under non-convex objectives, demonstrating how hierarchical momentum affects asynchronous training. Empirical results show HALoS achieves up to 7.5× faster convergence than synchronous methods and faster than existing asynchronous methods in geo-distributed LLM training, while preserving model quality.

**Claims And Evidence:**

Claim1: HALoS enables efficient geo-distributed LLM training.
Evidence1: experiments show the reduced amount of communication and enhanced speed.
Claim2: HAloS has tight convergence.
Evidence2: The proof in Appendix E.

**Essential References Not Discussed:**

No (from my knowledge).

**Experimental Designs Or Analyses:**

Experiments are carried out in a simulation manner. The designs are okay with four regions, 16 workers for LPS. Worker speeds are not constants; rather, they are randomly set via uniform distribution.

**Methods And Evaluation Criteria:**

I think the methods make sense. HALoS could reduce inter-region communication.
Benchmarks (MMLU, Hellaswag, LAMBADA, and WinoGrande) are commonly-used ones in LLM evaluation.

**Other Comments Or Suggestions:**

N/A

**Other Strengths And Weaknesses:**

Strengths:
1. I think the research field is of practical usages. During pretraining of LLMs, cross-region training is common.
2. The experiments are sufficient.

Weaknesses:
1. The introduction needs to be polished. It is much longer than I expected, and I think some chunk of paragraphs could be re-allocated to "related work".
2. Too much hyperparameters are introduced.

**Questions For Authors:**

Since many hyperparameters are introduced in this work, how to effectively adjust new hyperparameters under a different scenario (different LLM models, different # of regions, etc.)?

**Relation To Broader Scientific Literature:**

On top of sync optimizers, the proposed HALoS targets cross region training scenarios via async training, and proposes an algorithm that reduces cross-region communication costs, yielding higher training speed. It is also claimed by the authors to be the first giving theoretical proofs on convergence.

**Theoretical Claims:**

Convergence analysis includes an order approximation of gradient expectation. I briefly go through Appendix E and found no significant problem.

---

> ### Author Rebuttal · Authors · 2025-04-01
>
> # [W1] Refining Introduction and Related Work
> We appreciate the suggestion to enhance the clarity and readability of our paper. We designed the introduction to provide sufficient motivation and context for HALoS by outlining both the challenges of geo-distributed LLM training and relevant prior work to help readers understand our design choices early. That said, we greatly value readability for the community and will carefully revise the introduction and related work sections in the final version to further improve their structure and ease of understanding.
>
> ---
> # [W2] Hyperparameter Search Strategy
> We recognize that effective hyperparameter tuning is crucial for the practical deployment of new optimization frameworks across diverse models and environments. To address this challenge, we combined rigorous theoretical analysis with empirical validation, enabling systematic and efficient hyperparameter tuning across different LLMs and geo-distributed setups.
>
> Our paper provides **a thorough theoretical analysis and valuable insights** that guide the identification of optimal hyperparameters. For example, our theoretical results (Section 4.2) demonstrate that local momentum ($\beta_l$) significantly stabilizes training due to relatively homogeneous updates within each region, while global momentum ($\beta_g$) requires moderation to effectively handle stale updates at the global level. Specifically, our theory suggested optimal momentum parameters ($\beta_l$=0.9, $\beta_g$=0.5), which we empirically confirmed through ablation studies (Section 5.2).
>
> To further reduce the hyperparameter search cost, we adopted **a scale-up strategy**. Initially, we conducted comprehensive hyperparameter sweeps on the smallest model (Pythia-70M) and subsequently transferred the best-performing settings to larger models (Pythia-160M and Pythia-410M). This approach consistently improved convergence efficiency, significantly lowering tuning overhead. Similar approaches have been recently validated by other studies as well ([1], [2]).
>
> Our results further indicate **strong generalization of these hyperparameters** across varying infrastructure conditions (e.g., inter- and intra-region bandwidths; see Section 5.3 of the paper) and different LLM families (e.g., Llama and Qwen; for related empirical results, please see the section "Empirical Validation on Different LLM Families" below), demonstrating their robustness.
>
> As a next step, we plan to integrate automated hyperparameter tuning techniques—such as Bayesian optimization or adaptive online adjustment—by combining our theoretical convergence bounds with runtime monitoring. We believe this will facilitate easier adoption of HALoS in diverse scenarios without excessive tuning overhead.
>
> [1] Arthur Douillard *et al*. DiLoCo: Distributed Low-Communication Training of Language Models. WANT@ICML 2024.
>
> [2] Weigao Sun *et al*. CO₂: Efficient Distributed Training with Full Communication-Computation Overlap. ICLR 2024.
>
> ---
> # Empirical Validation on Different LLM Families
> To demonstrate the robustness and practical applicability of HALoS across diverse model architectures, we extended our evaluation beyond the Pythia models to include two additional widely-used LLM families—Llama and Qwen.
>
> |Model|Training Speedup of HALoS (vs. Async-Local-SGD)|
> |-|-|
> |Llama-70M|2.1x|
> |Qwen-70M|2.3x|
> |Pythia-70M|1.9x|
>
> As demonstrated in the table above, **HALoS consistently outperforms the strongest baseline (Async-Local-SGD) across all tested LLM architectures**. Importantly, the hyperparameters initially optimized for Pythia-70M—**without any additional tuning**—exhibited strong generalization, achieving even greater relative improvements (**up to 2.3x** with the Qwen-70M model).
>
> **Experiment Details:** We assessed whether the optimal hyperparameters, identified from our theoretical analysis and validated on the Pythia-70M model, could effectively generalize to these models. To ensure a fair comparison, we selected Llama and Qwen models closely matching Pythia-70M in size, using the same number of layers and hidden dimensions, and conducted the same experiments described in Section 5.1 of the paper.

---

### Official Review · Reviewer_eTdm · 2025-03-14

**Overall Recommendation:** 4

**Summary:**

The paper presents HALoS, an optimization framework designed to enhance cross-region training of large language models (LLMs). To address the challenges of communication costs and imbalanced hardware utilization, HALoS employs a hierarchical architecture with local parameter servers within each region and a global parameter server that aggregates updates across regions. By prioritizing fast intra-region communication and minimizing inter-region synchronization, the framework reduces latency, mitigates straggler effects, and optimizes resource efficiency. HALoS enables asynchronous updates at both local and global scales, facilitating overlap of computation and communication. The authors provide theoretical convergence guarantees for non-convex objectives, analyzing how hierarchical momentum influences training stability in asynchronous settings. Empirical evaluations demonstrate HALoS’s superiority.

**Claims And Evidence:**

HALoS could greatly improve training speed under cross-region scenarios, and it ensures convergence. Both experiment results and proofs in the appendix have verified the claim.

**Essential References Not Discussed:**

Not at all.

**Experimental Designs Or Analyses:**

Experiment designs are generally okay under the simulation regime. Randomness is introduced in terms of worker speed, which reflect real scenarios. But I think real experiment demos rather than simulations could be carried out to directly demonstrate the efficacy of the method.

**Methods And Evaluation Criteria:**

The methodology is okay with a good motivation and experiment-verified results.
The work uses common LLM benchmarks. There is no problems in terms of benchmarks.

**Other Comments Or Suggestions:**

No other comments.

**Other Strengths And Weaknesses:**

Strengths:
1. Practicalness of the proposed method; good simulations that reflect reality.
2. Theoretical proofs on convergence.
3. Scalability across various model sizes and network bandwidth conditions.

Weaknesses:
1. The proposed method assumes i.i.d. data distribution across all workers.
2. Global parameter server may undergo huge computation burden, impacting efficiency.

**Questions For Authors:**

How would the method perform under non iid? Would the method work without the iid assumption?

**Relation To Broader Scientific Literature:**

The work is contributive to the broader literature in that it has a hierarchical architecture: global & local. It also demonstrates good efficiency.

**Theoretical Claims:**

The convergence analysis is supported by a proof om the supplementary materials.

---

> ### Author Rebuttal · Authors · 2025-04-01
>
> # [W1] HALoS without i.i.d. assumption
> We appreciate the reviewer’s insightful question regarding the i.i.d. assumption. While our theoretical analysis in Section 4.2 follows standard practice by assuming i.i.d. data for analytical clarity, **HALoS does not rely on this assumption in practice.** To demonstrate this, we evaluate HALoS under both i.i.d. and non-i.i.d. data distributions.
>
> For the non-i.i.d. setting, we use the Shakespeare dataset [2], assigning each worker distinct speaking roles (e.g., Juliet, Romeo)—a widely adopted setup in federated learning (e.g., [1], [2], [3]). The i.i.d. setting is created by randomly shuffling data across workers. After training on the same amount of data, we report accuracy and relative training time below (see “Experiment Details” at the end for additional information):
>
> |Method|Test Accuracy|Relative Training Time|
> |-|-|-|
> |Async-Local-SGD|49.2%|1.6|
> |HALoS (w/ Global Momentum)|46.9%|1.0|
> |HALoS (w/o Global Momentum)|**50.0%**|**1.0**|
>
> Even under non-i.i.d. partitions, **HALoS (w/o Global Momentum) achieves 0.8% higher accuracy and 1.6× faster convergence** than our strongest baseline, Async-Local-SGD. As discussed in Section 4.2, global momentum may slow convergence when worker updates conflict due to data heterogeneity, while local momentum remains effective. This allows HALoS to mitigate inter-region communication delays and converge efficiently.
>
> In the i.i.d. setting, HALoS achieves the same accuracy (50.4%) as Async-Local-SGD but with 1.6× faster training. These results highlight **HALoS’s robustness across both i.i.d. and non-i.i.d. settings,** demonstrating practical effectiveness beyond theoretical assumptions.
>
> The slight accuracy drop under non-i.i.d. data (50.0% vs. 50.4%) is consistent with prior observations in many hierarchical or asynchronous methods, which are *partially robust* to data heterogeneity. To further improve non-i.i.d. performance, we plan to explore FL techniques such as gradient clipping and adaptive regularization ([3]).
>
> Theoretically, **HALoS can be readily extended to non-i.i.d. settings**. Existing literature on local or federated SGD often provides additional bounds involving the divergence across local distributions. In HALoS, we can introduce an extra term in the bound related to the data heterogeneity assumption when bounding the variance of local gradients, similar to prior work ([4], [5]). This would allow our analysis to formally reflect the effect of data heterogeneity.
>
> We thank the reviewer again for prompting this valuable discussion. We will include both the empirical results and an extension of our theoretical analysis to the non-i.i.d. case in the final version.
>
> **Experiment Details:** We train a 6-layer Llama-style model (128 hidden dim) to predict the next character (among 79 unique characters) given the previous 80. We train on 3,145,728 samples with a batch size of 64 per worker using 16 workers in Appendix A. Evaluation uses 524,288 separate test samples. We use a max local learning rate of 0.01 and train for one epoch. For HALoS, we use the same hyperparameters in our paper. For Async-Local-SGD, we found that 32 local steps ($H$) caused divergence and adjusted it to $H=16$.
>
> [1] Brendan McMahan et al. Communication-Efficient Learning of Deep Networks from Decentralized Data. AISTATS 2017.
>
> [2] Sebastian Caldas et al. LEAF: A Benchmark for Federated Settings. arXiv preprint arXiv:1812.01097, 2018.
>
> [3] Junbo Li et al., FedNar: Federated Optimization with Normalized Annealing Regularization, NeurIPS 2023.
>
> [4] Xiang Li et al., On the Convergence of FedAvg on Non-IID Data, ICLR 2020.
>
> [5] Tian Li et al., Federated Optimization in Heterogeneous Networks, MLSys 2020.
>
> ---
> # [W2] Computation in Global Parameter Server (GPS)
> We acknowledge that minimizing computation in the Global Parameter Server (GPS) is crucial for scalable geo-distributed training, and this goal fundamentally guided our design of HALoS. Our method significantly reduces GPS computation by employing local server-side update accumulation: each local parameter server (LPS) aggregates multiple local updates before synchronizing with the GPS. This design reduces both the frequency of inter-region communication and the number of updates processed by the GPS—preventing it from becoming a bottleneck.
>
> |Method|# Global Updates|Time-to-Loss|Tokens-to-Loss|
> |-|-|-|-|
> |FedAH|5857|3.3|38.1B|
> |HALoS (Ours)|478|1.0|12.3B|
>
> As shown in the above table (extending Table 2 in the paper), when training Pythia-70M, **HALoS reduces GPS updates by 12.3x** compared to FedAH, our baseline hierarchical asynchronous training algorithm. This reduction comes from two key factors: (i) 32 local update accumulations per LPS before GPS communication, and (ii) 3.1× fewer total tokens needed to reach the same loss due to HALoS’s improved efficiency. Together, these design choices substantially reduce GPS computation and ensure HALoS achieves scalable, efficient coordination.

---

### Decision · Program_Chairs · 2025-05-01

**Decision:**

Accept (poster)

**Comment:**

This submission presents HALoS, a hierarchical asynchronous optimization framework for geo-distributed LLM training, addressing communication bottlenecks and heterogeneous hardware challenges. The reviewers initially raised concerns about non-i.i.d. data robustness, GPS computational overhead, hyperparameter tuning, and generalizability across heterogeneous clusters and model architectures. The authors provided thorough rebuttals, including empirical validation of HALoS’s performance under non-i.i.d. settings, demonstrating reduced global update frequency, and extending evaluations to other models. Theoretical arguments were also extended to account for data heterogeneity. Reviewers acknowledged these clarifications, with the unanimous positive reviews. All reviewers ultimately support acceptance.